

# 1 Semi-automatic sunshots with the WIDIF DIflux

Jean L. Rasson, Olivier Hendrickx, Jean-Luc Marin
Royal Meteorological Institute, Centre de Physique du Globe, Dourbes, 5670, Belgium
*Correspondence to*: Jean L. Rasson (jr@oma.be)
**Abstract.** The determination of magnetic declination angle entails finding two directions : Geographic North and
Magnetic North. This paper deals with the former. The known way to do it by using the Suns` calculable orientation in
the sky is improved by using a device based on a WIDIF Diflux theodolite and split photocells positioned on its telescope
ocular. Given the WIDIF accurate timing and localisation provided by the on board GPS receiver, an astronomical
computation can be effectuated and the Sun azimuth as well as an auxiliary's target azimuth can easily and quickly be
determined. The precise Sun`s crossing of the split photocell, amplified by the telescope's magnification allows
azimuth` accuracies of a few seconds of arc.
1 True North
The determination of True North via the mark's azimuth required for magnetic declination is an old problem which has
received a number of solutions (Šugar et al., 2013): by Sunshot, North seeking gyroscope (Rasson & Gonsette, 2016),
GPS techniques (Lalanne, 2013).
The sunshot technique, although it is potentially quick and accurate is not very popular. The reason probably stems from
fear of suffering eye damage when trying to point the sun with a telescope, the supposed difficulty of astronomical
computations and of course the impossibility of carrying on sun observations in cloudy weather.
The sunshot technique is not cumbersome and basically needs only few types of equipment:
1) A theodolite with adequate precision. This can be the very Diflux theodolite used for magnetic measurements,
2) Diagonal eyepieces (ocular and microscope) adapted for the theodolite in use so that steep sightings can be carried
out (in case of high Sun elevation),
3) A solar filter fitting on the eyepiece ocular,
4) Precise time and localisation (WGS84 lat/long),
5) Conversion data from UTC to UT1 and
6) Astronomical tables or software for the Sun ephemerides of the current year.
An accuracy of about 1 arcsec in the azimuth can be achieved in the best cases, that is if:
• Timing is better than 0.1s in UT1
• The latitude and longitude of the sunshot determination place is known to better than 1 arcsec in the WGS84 datum

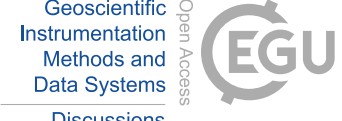



• The theodolite leveling is achieved to better than 5" and the sunshots are performed at the time of sunrise or sunset
when the sun has a low elevation. Leveling with this precision is quite possible, even on a tripod but frequent level
checks and adjustments are required. An idea of the errors associated with leveling is given on Figure 1.
The sunshot technique has notable advantages: it is rather easy to use, does not need much additional equipment, requires
the occupation of a single station only and is fast: an experienced observer will not take more than a few minutes of time
per sunshot.
## 2 First test set-up for semi-automatic sunshots
In our attempts to automate sunshots we tried all kinds of set-ups. For instance for a rather rough set-up we isolated a
sunray as sunlight passing through a pinhole in a black box. This ray was reflected back by a shiny sphere to amplify the
horizontal angular motions of the ray, which then fell onto a split photocell (2 horizontally side-by-side cells). The useful
signal was therefore the difference voltage produced by both illuminated photocells. We had to observe this changing
signal with the apparent Sun motion: the set-up pointed to the Sun when the difference voltage was zero.
## 3 Theodolite's telescope with split photocells
Another set-up – and the final one – makes use of the theodolite's telescope itself. Indeed the optical magnification is
usually ~30x and can be used efficiently by installing the split photocells in front of the ocular side-by-side. Additionally,
if the theodolite is of the WIDIF Diflux type pictured on Figure 2 (Rasson et al. 2016), a GPS receiver for precise time
and positioning is available. We concluded it was necessary to construct a small mechanical and electronic sunshot add-
on able to be slid and fastened on the theodolite's ocular. Electrical and computing power could then be borrowed from
the WIDIF itself.
**3.1 Mechanical and electronic design**
Such an add-on has been designed and manufactured and is displayed on Figure 3. The Sunshot add-on is clamped on the
telescope at the ocular end. An articulated cover, holding the photocells can be opened so as to leave the ocular
accessible for normal telescope pointing with the eye. The cover holds two photocells plus some analogue electronics in
front of the ocular lens (when closed) so as to catch the light passing through the telescope and transform it into voltages.
The add-on generates "SUM" and "DIFFerence" of the two photocell signals. A small cable runs from the add-on and
can be plugged into the WIDIF input/output connector also used for WIDIF battery charging (Figure 4). The voltages are
displayed on the WIDIF LCD screens in numerical form (Figure 5).



## 3.2 Add-on readings for sunshot

The procedure for performing a sunshot uses the apparent horizontal motion of the sun moving its sunrays through the telescope. The focussed rays will sweep over both cells and stop the clock timer when the difference between the 2 photocell voltages is zero. But a zero will also exist if no light at all falls on the photocells. Therefore we inspect also the SUM of the photocell voltages:

If SUM > 600, it means that the Sun illuminates both photocells and

If at the same time DIFF=0 the telescope axis points to Sun; we have a valid Sunshot.

We can now put together a semi-automatic sunshot procedure for a WIDIF theodolite equipped with a sunshot add-on:

1. Point the telescope axis towards the Sun with the theodolite's vertical circle to the right (CR)
2. Use "SUM" signal by maximizing it to centre the Sun's image on the photocells using the theodolite's H & V slow-motion screws
3. Using H slow-motion screw, point slightly ahead of he Sun (to the right of Sun in Northern hemisphere) so that "DIFF" is about 100.
4. Start the zero-crossing detector by depressing the service switch on the WIDIF
5. The Earth rotation moves the Sun image on the photocells
6. When DIFF=0 zero crossing occurs, the clock is automatically stopped and the UTC time is displayed
7. Read time and read the HC
8. Convert from UTC to UT1
9. Compute Sun's azimuth
10. Repeat with CL

As an example of the capabilities of this sunshot add-on working with a WIDIF theodolite, we performed the measurement of the azimuth of the Mark as seen from the D05 new pillar installed for the Diflux intercomparison session during the Instruments IAGA Workshop in Dourbes during August/September 2016. Results are given in Figure 6 where the UTC-UT1 correction has been applied. We can appreciate the low dispersion of the results and the rather stable values over time.

## 4 Special precautions to improve the azimuth accuracy

The time provided by GPS receivers is usually UTC. The difference between UTC and UT1 is due to Earth rotation irregularity and is kept below 1 s. This translates to a maximum of about 10 arcseconds in the Sun's azimuth. So for an accuracy beyond that, the correction to UT1 should be applied. It is available on this website: http://maia.usno.navy.mil/ser7/ser7.dat

Since the WIDIF has a reading resolution of 1 arcminute, which can be interpolated to 0.1 arcminute by eye, it is good practice to preset the index on the HC at existing marks of 1.0 minutes in step 3. No interpolation is then necessary,




eliminating any uncertainty associated with it.
Levelling is quite critical for a sunshot and the more so when the Sun has high elevation (Figure 1). Therefore a preferred
time for maximizing the accuracy is sunrise or sunset with the Sun is low over the horizon.
**5 At or near the Equator**
At or near the Equator the Sun has no or little horizontal motion.  To obtain a zero-crossing from the photocells, it is then
necessary to rotate the theodolite around its vertical axis.
Therefore the HC slow-motion screw must be used to manually trigger the zero-crossing detector. This may degrade the
accuracy as the operator may overshoot the zero-crossing. It may be better to operate manually in those equatorial
conditions (see below).
**6 Further developments**
Provision has been made to perform the astronomical calculation of the Sun's azimuth inside the WIDIF electronics,
using the epoch and Lat/long information collected by its GPS receiver at the time of the sunshot. The algorithm used for
the computation is the one provided in the Guide for Magnetic Repeat Station Surveys programmed by Andrew Lewis of
Geoscience Australia (Jankowski and Sucksdorff 1996, Newitt et al. 1996) and does not need the input from an
Astronomical Almanac. The computation results provided by this algorithm have been checked to be correct within 2
arcseconds by comparison with a master program providing sub arcsecond accuracy (Reda and Andreas 2003).
The WIDIF will also be upgraded in order to perform the sunshots manually, without the add-on being necessary. The
photocells zero crossings epochs will then be logged manually instead by depressing the WIDIF service switch. The
timing by hand may not be as precise as the one provided by the photocells, except at the equator.

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

Competing interests
The authors declare that they have no conflict of interest.


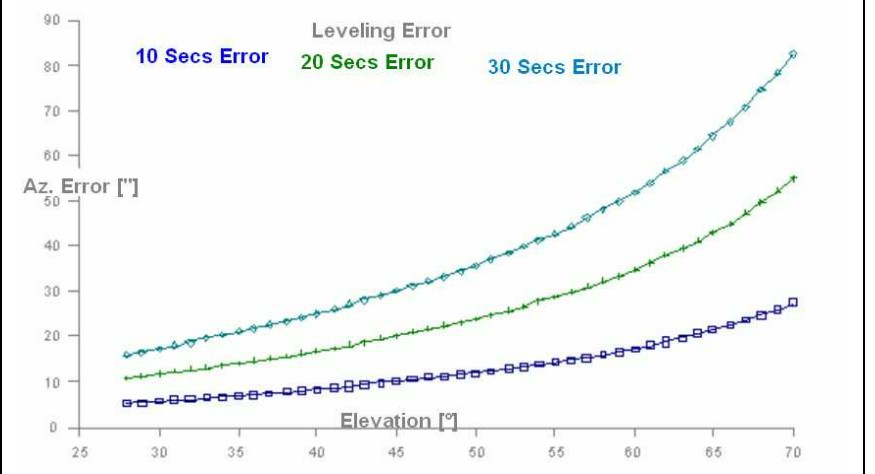


**Figure 1. Sun's azimuth error associated with 3 theodolite's leveling errors vs the Sun's elevation**





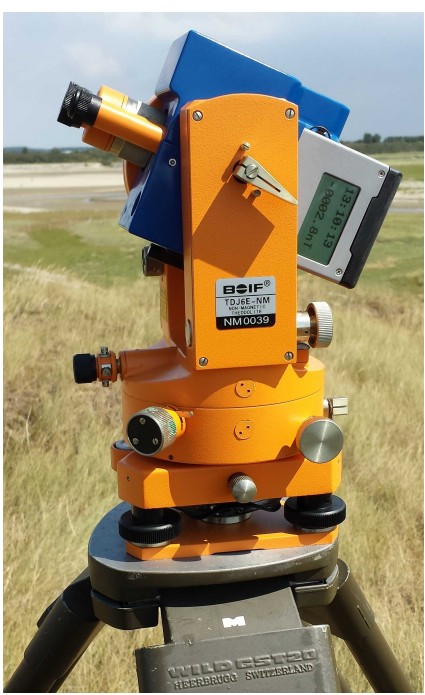

**Figure 2. The WIDIF DIflux theodolite packs fluxgate sensor and electronics as well as GPS receiver and a battery on the**
**telescope.**

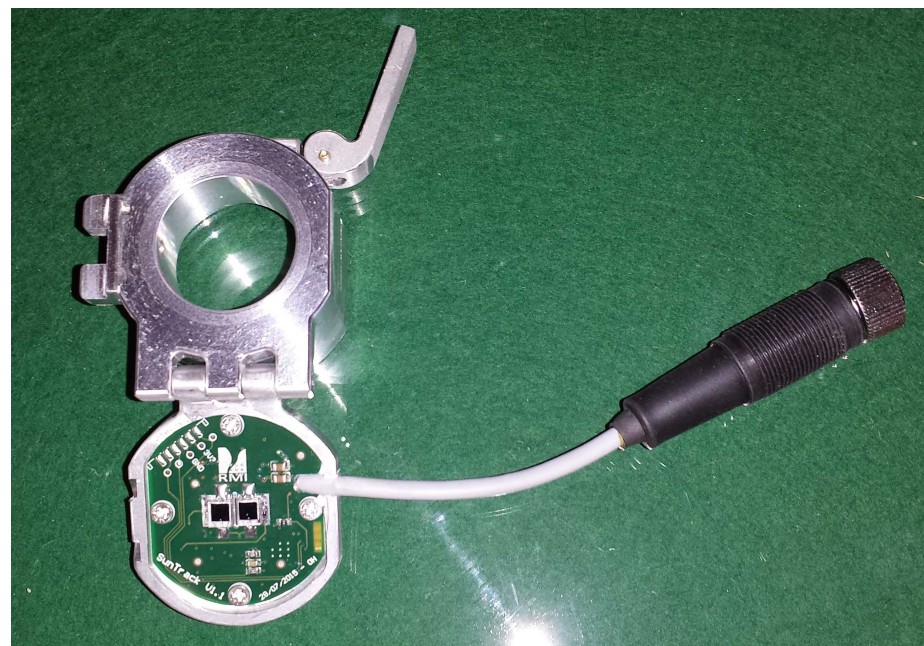

**Figure 3. The sunshot add-on is a small mechatronic device fitted to the WIDIF theodolite's telescope ocular and able to**
**precisely determine the Sun's direction. Two black photocells serving this purpose can be seen side-by-side on the printed**
**circuit board.**





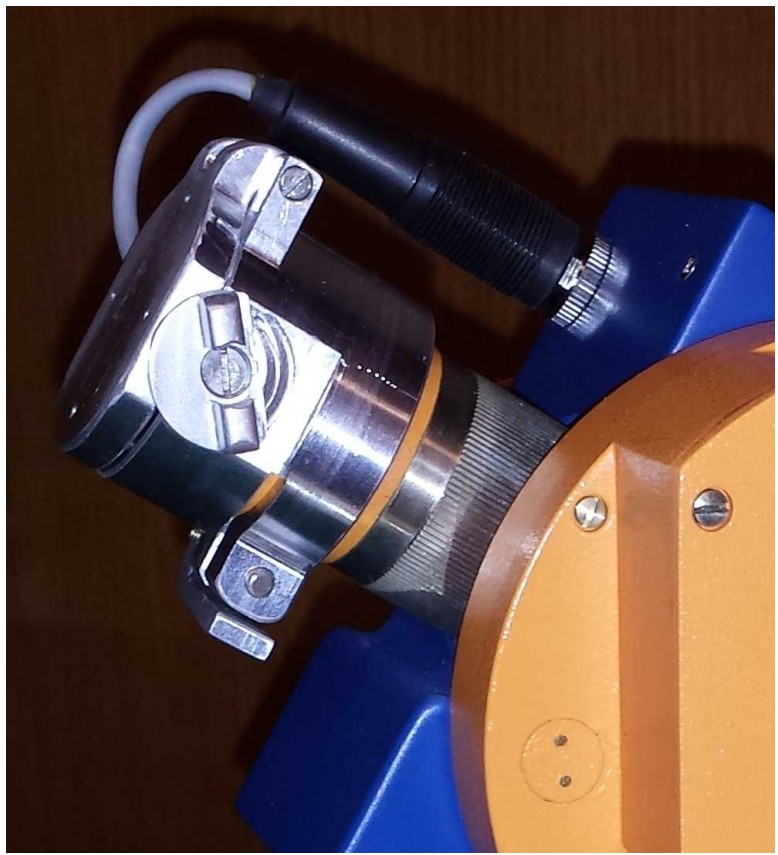

2 **Figure 4. The sunshot add-on mounted on the WIDIF telescope. The device draws its power from the WIDIF battery .**

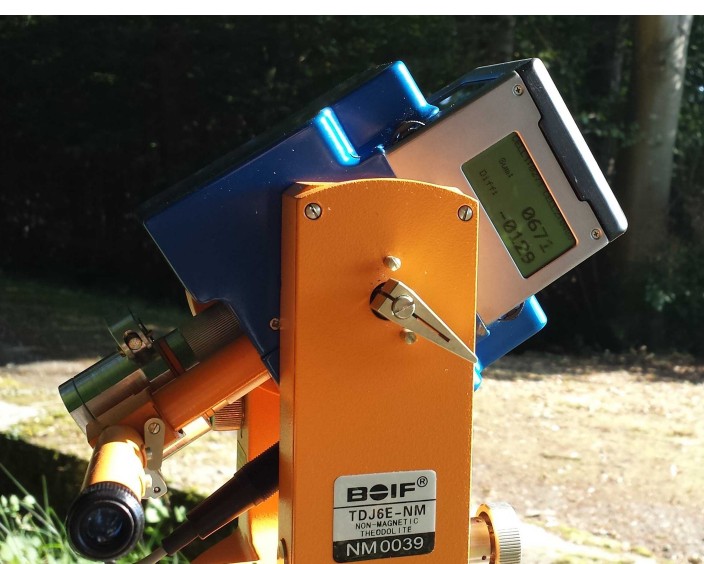

4 **Figure 5. The photocells sum and difference voltages are displayed on the WIDIF LCD's so as to ascertain the way the cells are**
5 **illuminated.**





| UT1=UTC-0.23s | Az. D05 from WS Mark | | |
|---|---|---|---|
| | ° | ′ | ″ |
| 17/08/2016 | 170 | 12 | 24.9 |
| 25/08/2016 | 170 | 12 | 24.6 |
| 2/09/2016 | 170 | 12 | 24.2 |
| | | | |

2 **Figure 6. Some results obtained with the Sunshot add-on for azimuth determination. Each azimuth listed is the mean of 10 CR**
3 **and CL independent sunshots.**