# Peer review of "Semi-automatic sunshots with the WIDIF DIflux"

_Geoscientific Instrumentation, Methods and Data Systems, 2017_

## Referee Comment (RC1)

**Reviewer comments to gi-2017-11-abstract-version 2**

The considered submission presents the design, development, and operational procedure of a split photocell attachment for the WIDIF theodolite. The system automates the timing measurements required when making sun observations in order to determine geodetic north. An azimuth accuracy to a few seconds' of arc is demonstrated.

The reviewer's recommendation is that the submission is published, on the proviso that the authors consider the following comments:

The authors are recommended to use the term 'position' or 'location' instead of 'localisation' (Page 1, Line 7; Page1, Line 24) and 'determination place' (Page 1, Line 29).

Page 1, Line 27: 'An accuracy of about 1 arcsec in the azimuth' – clarify if this accuracy refers to the sun azimuth being measured, or the final geodetic north azimuth calculated.

Page 2, Line 1: New bullet point for 'sunshots are performed… …low elevation.' ?

Page 2, Line 5: 'few minutes of time per sunshot'? Clarify if this is the surveying time only, does this include carrying out the calculations?

Page 2, Line 16: (Rasson et al. 2016) should refer to 2016a or 2016b as per References section.

Page 3, Line 1: A title like 'Sunshot Readings from Add-on' would more specifically refer to the output results from the device.

Page 4, Line 3: '…with the Sun  low over the horizon'

Page 8, Figure 6 (General comment): The results appear very consistent. The author could consider including figures to show how the accuracy/repeatability of results from the add-on compares to manual sunshots, and another geodetic north azimuth method value as a reference.

---

## Referee Comment (RC2) · Anonymous Referee #2 · 12 Apr 2017

Semi automatic sunshots with the WIDIF DIFlux Jean L. Rasson, Olivier Hendrickx, Jean-Luc Marin

Overall Quality This paper presents a useful and novel method to semi-automate the determination of azimuth using the azimuth-by-hour-angle method of sun observation with a photo-cell electronic add-on mounted on the telescope of a declination-inclination fluxgate theodolite.

I recommend the paper be published with minor additions and changes.

Individual Scientific Questions The method described in the paper uses a zero difference in the output of the two photo-cells and the observing method calls for observations with vertical circle right and vertical circle left, presumably to correct, via averaging, for photo-cell collimation error or unbalanced output voltage from the individual

photo cells. I think the paper would be improved if this aspect was briefly discussed explicitly.

The discussion on page 4, line 17, 18 and 19 requires further explanation. I interpret the statement on line 17 "without the add-on being necessary" to mean the photo-cell attachment is not connected to the theodolite so how is it then possible to manually time photo-cell zero crossing epochs?

I would be interested to see the standard deviation of the individual readings used to calculate each of the three average azimuths included in table 6.

Technical Corrections There is inconsistency in capitalisation throughout the paper and some minor grammatical corrections required. The paper refers indirectly to an algorithm to calculate the location of the Sun but should specifically reference the paper Bennet, G.G."A solar ephemeris for use with programmable calculators", The Australian Surveyor, Vol. 30, No. 3 pp 147-151

Please see the accompanying PDF file containing annotations for specific details.

Please also note the supplement to this comment:
http://www.geosci-instrum-method-data-syst-discuss.net/gi-2017-11/gi-2017-11-RC2-supplement.pdf

——————————————————

[Figure]

**Supplement:**

[revised manuscript text omitted]

---

## Author Comment (AC1) · 30 May 2017

Please find here the revised manuscript (v 2.0) taking into account the comments from both reviewers. Many thanks to them for the in depth review and very useful comments

Please also note the supplement to this comment:
http://www.geosci-instrum-method-data-syst-discuss.net/gi-2017-11/gi-2017-11-AC1-supplement.pdf